# Breastfeeding Duration, Diet, and Sports Engagement in Immigrant Children: A Quantitative Study in the Lisbon Region, Portugal

**DOI:** 10.3390/nu17081350

**Published:** 2025-04-15

**Authors:** Zélia Muggli, Regina Loesch, Iolanda Alves, Iliete Ramos, Maria Rosario O. Martins

**Affiliations:** 1Global Health and Tropical Medicine, Institute of Hygiene and Tropical Medicine, NOVA University of Lisboa, 1349-008 Lisbon, Portugal; zelia.muggli@ihmt.unl.pt (Z.M.); regina.loesch@ihmt.unl.pt (R.L.); iolandaa@ihmt.unl.pt (I.A.); 2ULS Arco-Ribeirinho, Unidade de Saúde Pública Arnaldo Sampaio, 2834-003 Barreiro, Portugal; iliete.ramos@ulsar.min-saude.pt

**Keywords:** immigrant children, fruit and vegetables intake, physical activity, breastfeeding practices, social determinants of health, migrant children in Portugal

## Abstract

**Background:** Being breastfed, following a healthy diet and staying active during childhood shape health trajectories across the life course, promoting long-term well-being. Despite the growing immigrant child population in Portugal, evidence on these behaviours, particularly among preschool-aged children, remains limited. In this context, this study examines the associations between migrant status and breastfeeding patterns, fruit and vegetable consumption, and participation in sports among children living in the Lisbon Metropolitan Area. **Methods**: A cross-sectional study nested within a cross-sequential cohort was carried out in the Lisbon region between May 2022 and April 2024. Approximately 760 children (49.4% immigrants) born in 2018 and 2020 were enrolled in the study. Data were collected through a face-to-face interview with parents, using a structured questionnaire with information on socioeconomic variables, migration history, breastfeeding habits, and children’s diet (fruit and vegetable consumption) and physical activity. We used parametric and non-parametric tests to compare migrant and non-migrant children. To quantify factors associated with the main outcomes, we estimated a logistic regression model and calculated crude and adjusted odds ratios and their respective 95%CI. **Results**: Immigrant children were disproportionately represented in socioeconomically disadvantaged families. Breastfeeding initiation rates were higher among immigrant mothers (96.3% vs. 87.6%, *p* < 0.001). Additionally, immigrant mothers had a longer median duration of any breastfeeding (14 vs. 8 months, *p* < 0.001) and of exclusive breastfeeding (6 vs. 4 months, *p* < 0.001). Immigrant children had significantly lower odds of consuming three or more portions of fruit (aOR = 0.700; 95%CI: 0.511–0.959; *p* = 0.027) and two or more portions of vegetables per day (aOR = 0.489; 95%CI: 0.350–0.684; *p* < 0.001) compared with non-immigrant children. They were also twice as likely to not engage in sports (aOR = 2.185; 95%CI: 1.512–3.158; *p* < 0.001). **Conclusions**: Breastfeeding was better implemented in immigrant children. The findings highlight the need to address social determinants of health and the challenges faced by immigrant families in promoting a balanced diet and sports participation for their children. Multisectoral, culturally appropriate interventions that sustain and promote good breastfeeding practices, improve access to healthy food, and encourage sports are crucial to reducing health inequalities.

## 1. Introduction

In recent years, significant increases in migration flows to Europe have been driven by conflict, economic instability, and environmental crises [1,2]. In 2023, 6% of people living in the EU were citizens born outside the EU [3], and many were children. In Portugal, the population of immigrant children is growing, with 29.2% of live births in 2023 involving foreign-born mothers living in the country [4]. The Lisbon Metropolitan Area recorded an even higher proportion in the same year, with 44.5% of live births involving mothers of foreign nationality, mainly from Brazil and African Portuguese-speaking countries [4].

Portugal has implemented progressive policies to support migrant integration, including access to health care and education [5]. Nonetheless, similarly to in other countries, migrant families often experience higher levels of poverty and food insecurity compared to native-born families, primarily due to socioeconomic disadvantages, cultural barriers, and difficulties in accessing essential resources in the host country [6,7,8,9,10]. Accordingly, immigrant populations face unique challenges that may affect their breastfeeding practices, dietary choices, and children’s overall health [10]. Breastfeeding is universally recognized as a critical component of infant nutrition and development, with established long-term benefits for children’s physical health, cognitive function development, and emotional well-being [11]. The World Health Organization (WHO) recommends exclusive breastfeeding for the first six months of life, followed by continued breastfeeding alongside the introduction of complementary foods up to at least two years of age [11]. However, global exclusive breastfeeding rates remain below these recommendations; from 2015 to 2020, only about 44% of infants aged 0 to 6 months worldwide were exclusively breastfed [12]. Results from the WHO European Childhood Obesity Surveillance Initiative (COSI) 2015/2017 indicated that in only one-third of the surveyed countries, more than 25% of children were exclusively breastfed up to six months of age. In Portugal, the prevalence was 21% [13]. Breastfeeding practices vary across cultures and are influenced by sociodemographic and economic factors. Some studies report differences in breastfeeding practices between migrant and native-born mothers, shaped by factors such as country of origin, length of stay in the host country, and acculturation [14,15,16,17,18,19]. Yet, a study conducted in Portugal found that the maternal country of birth does not affect breastfeeding initiation and that breastfeeding duration remains consistent regardless of how long the mother has lived in the country [20]. Additionally, studies suggest that immigrant mothers generally have higher breastfeeding initiation rates [21] and a longer breastfeeding duration, whether exclusively or in any form, compared with native-born mothers [20,22]. In contrast, a study from Denmark found that non-Nordic migrants and their descendants were more likely to have suboptimal breastfeeding than women of Danish origin [23].

Children of immigrant parents more often experience an imbalanced diet [24] and an unhealthy lifestyle [25], which can prevent them from reaching their full potential, and may reinforce health disparities and social inequalities [26]. Worldwide, undernutrition is associated with 45% of child deaths [12]. Moreover, migrant children in European countries have a higher risk of malnutrition [10]. Fruit and vegetable consumption serves as a key indicator for assessing diet quality. In 2015–2016, a national survey from Portugal found that 72% of children had insufficient consumption of fruits and vegetables [27]. The Portuguese Health Directorate (DGS) recommends a daily intake of 2–3 portions of fruit and 3–4 portions of vegetables per day for 1–6 year olds [28]. Proper nutrition during early childhood, including sufficient consumption of fruits and vegetables, lays the foundation for lifelong health by helping to reduce morbidity and mortality, lowering the risk of non-communicable diseases, supporting overall development, and significantly influencing long-term health outcomes [29,30,31]. Studies show that migrant status is commonly associated with food insecurity, with rates reported in this group being higher than those in the general population, and with a higher risk of low vegetable/fruit consumption in children [8,9,32,33,34,35,36]. In contrast, a study in the Netherlands found that migrant children consumed more fruits and vegetables than their native peers [37].

Physical activity (PA) intersects with breastfeeding and dietary habits for healthy child development. It enhances overall children’s health by promoting bone strength, healthy growth, and muscle development, and improving motor and cognitive skills, while reducing the risk of childhood obesity and related non-communicable diseases later in life. Moreover, PA promotes better mental health and well-being [38,39]. Research suggests that immigration background can pose a barrier to PA among children, as immigrants more often exhibit lower levels of physical activity than non-immigrant children [40,41]. This may be attributed to reduced access to safe recreational spaces, acculturation challenges, and socioeconomic constraints [42,43]. Organized sports offer significant benefits for increasing physical activity levels, yet studies indicate that immigrant children participate less frequently [44,45,46].

In Portugal, most studies on the migrant population have focused on adults, while evidence on the nutritional behaviour and physical activity of preschool-aged immigrant children remains scarce. In this context, this study aimed to analyze the associations between migrant status and breastfeeding duration, fruit and vegetable consumption, and participation in sports in immigrant and non-immigrant children living in the Lisbon Metropolitan Area in Portugal.

## 2. Materials and Methods

This is a cross-sectional study nested within a cohort study. The original study followed a cross-sequential cohort of children born in 2018 and 2020 in the Lisbon region, Portugal. In this paper, we use data from the first wave of data collection from each age cohort.

### 2.1. Study Setting

The study was carried out in four municipalities in the Lisbon region of Portugal (Barreiro, Moita, Montijo, and Alcochete). In 2022, the region included 17 primary health care centres with around 240,000 users, of whom 3.4% were children aged between 1 and 5 years. The region also has two hospitals.

### 2.2. Participants

We included children born in 2018 and 2020, living in the four municipalities and attending primary health centres. There are no figures on child users disaggregated by immigrant status, but we know, for example, that in 2023, in one of the largest public maternities in Lisbon (Maternidade Alfredo da Costa), more than 40% of children were born to a foreign mother. We defined children as immigrant if they resided in Portugal and were born outside the EU, or had at least one parent born outside the EU. We included in our study all immigrant children attending health centres during the period of recruitment. Primary health centres were randomly selected, and one non-immigrant child was matched to one immigrant, to overcome possible loss to follow-up of immigrants.

### 2.3. Recruitment

We recruited children based on their health and vaccination appointment schedules at primary health care centres in their area of residence. First, we selected children born in 2018 and 2020 from these schedules. Then, at the scheduled time, we approached all parents/caregivers who attended a consultation or vaccination appointment with their child in the waiting room. Recruitment took place sequentially, with children born in 2018 and 2020 enrolled between May 2022 and April 2024.

### 2.4. Data Collection

#### 2.4.1. Socioeconomic, Demographic, and Children’s Health Variables

At baseline (i.e., the first wave of data collection for each age cohort), we conducted face-to-face interviews with the child’s parents/caregivers, using a questionnaire to collect socioeconomic and demographic data. The interviews took place in a private area within the primary health care centre to ensure complete privacy. Our research team included immigrant recruiters/interviewers with diverse backgrounds who had received specialized training to ensure accurate data collection. The interviews were conducted in Creole, Brazilian Portuguese, and English, in addition to Portuguese, as needed. The study examined various variables, including the caregiver’s age, sex, educational level, employment status, perception of the child’s health status, monthly household net income, family structure, the child’s migrant status and migration history, health insurance, and access to health care. We also collected information on children’s diet (fruit and vegetable consumption) and physical activity, namely participation in organized sports.

#### 2.4.2. Main Outcomes

We analyzed two main outcomes: dietary habits and participation in organized sports. Dietary habits were divided into fruit and vegetable consumption: a binary variable equal to one if fruit consumption was above the median, and zero otherwise; and a binary variable equal to one if vegetable consumption was above the median, and zero otherwise. For participation in organized sports, the main outcome was one if the children did not engage in organized sports, and zero otherwise.

The secondary outcome was related to breastfeeding practices, and was measured as the median duration of any breastfeeding and the median duration of exclusive breastfeeding.

### 2.5. Statistical Analysis

Descriptive statistics were used to characterize the sample variables. The two-sample z-test was used to compare proportions between groups, and the Mann–Whitney U test was used to compare medians. Associations between variables were analyzed using Qui-Square tests (Fisher and Fisher–Freeman–Halton Exact Tests). Odds ratios and corresponding 95%CIs were estimated using logistic regression. We set the significance level at 5% and performed the statistical analysis in SPSS software version 28.

## 3. Results

### 3.1. Children and Families Sociodemographic Characteristics

A total of 760 children participated in the study, with 55% (*n* = 422) born in 2018 and 45% (*n* = 338) born in 2020. Data collection occurred when the children were between the ages of three and five. There was an equal distribution between non-immigrant (*n* = 380) and immigrant children (*n* = 380), due to the matching procedure. The number of children analyzed differed for each variable, as not all participants responded.

The primary caregivers of the immigrant children originated from 27 countries, predominantly from the Community of Portuguese-Speaking Countries (CPLP). The most represented countries included Brazil (22.2%), Angola (21.4%), Cape Verde (17.9%), and Guinea-Bissau (14%), while other immigrant origins included countries from the Indian subcontinent (4%). The median length of stay in Portugal for the mothers of immigrant children was five years.

The main sociodemographic characteristics of the children and their families can be found in Table 1.

Significant differences were observed in both maternal education (*p* = 0.003) and employment status (*p* < 0.001) between immigrant and non-immigrant children. Among mothers of immigrant children, 50.7% had secondary or professional education, compared with 38.7% of their counterparts. In contrast, mothers of immigrant children (29% vs. 38.4%) were less likely to have completed university education compared with mothers of non-immigrant children (29% vs. 38.4%). Employment status also varied significantly between the two groups, with a higher proportion of mothers of immigrant children (43.0%) being unemployed, retired, or students, compared to 29.8% of mothers of non-immigrant children. Income levels further highlighted socioeconomic disadvantages among immigrant children. Immigrant families (34.3%) were one and a half times more likely to have a very low monthly income of up to €750, compared to families of non-immigrant children (22.9%). Nearly half of non-immigrant families (47.6%) earned more than €1500 per month, whereas two times fewer immigrant families (20.4%) fell into this category.

In addition to the socioeconomic profile of disadvantage observed among immigrant children, the data also suggest the presence of inequalities in health care access and utilization. Fewer immigrant children had an assigned family doctor compared to non-immigrant children (54.1% vs. 81.3%; *p* < 0.001). Moreover, immigrant children had lower attendance rates at age-targeted health and development checks. At age three, 58.8% of immigrant children born in 2020 were assessed, compared to 75.0% of their counterparts (*p* = 0.003). Similarly, among children born in 2018, 45.7% of immigrant children versus 57.6% of non-immigrant children were examined at age 3 (*p* = 0.019).

### 3.2. Breastfeeding, Fruit and Vegetable Intake, and Associated Factors

As shown in Table 2, dietary patterns differed significantly between the two groups.

Regarding breastfeeding, data analysis indicated that mothers of immigrant children were not only more likely to initiate breastfeeding (96.3% versus 87.6%; *p* < 0.001), but were also more likely to exclusively breastfeed for a longer duration (median: 6 vs. 4 months; *p* < 0.001). Furthermore, the total duration of any breastfeeding was longer among immigrant children, with a median of 14 months, whereas mothers of non-immigrant children breastfed for a median of 8 months.

Over 80% of children in both groups met the daily recommended fruit intake (≥2 portions/day) for this age group. However, significant differences in the frequency of intake were observed between immigrant and non-immigrant children (*p* < 0.001). A greater proportion of non-immigrant children (55.8%) consumed three or more portions per day, compared to 45.6% of immigrant children. In contrast, a higher percentage of immigrant children (13.8%) consumed less than two portions per day, compared to 10.5% of non-immigrant children. In stark contrast to the high proportion of children meeting the recommended daily fruit intake, fewer than 20% of children met the recommended daily vegetable consumption (≥3 portions/day), with similar percentages between immigrant (16.9%) and non-immigrant children (14.7%). Nevertheless, distinct vegetable consumption patterns emerged between the groups (*p* < 0.001). The proportion of children with zero or very low intake (one portion/day) was lower among non-immigrant children (30% vs. 45%), while the intake of two portions/day was more prevalent among non-immigrant children (55.3%) when compared with immigrant children (37.8%).

Logistic regression analysis was performed to identify the factors associated with daily intake above or below the median, defined as three or more portions of fruit and two or more portions of vegetables (Table 3 and Table 4). The results suggest that, after adjusting for other factors, immigrant children had 30% lower odds of consuming three or more portions of fruit per day compared to non-immigrant children (aOR = 0.700; 95%CI: 0.511–0.959; *p* = 0.027). Additionally, both immigrant status (aOR = 0.489; 95%CI: 0.350–0.684; *p* < 0.001) and belonging to the 2018 birth cohort (aOR = 0.550; 95%CI: 0.398–0.781; *p* < 0.001) were associated with reduced odds of consuming at least two portions of vegetables per day. The findings also suggest that variables related to maternal education and family income did not show a significant association with fruit or vegetable intake in either model.

We also used the subset of children with a migrant background (*n* = 360) to analyze whether length of stay in the host country was associated with nutritional patterns. After adjusting for family income and maternal education level, the length of stay in the host country among immigrant mothers was not associated with fruit intake above the median (aOR = 1.016; 95%CI: 0.990–1.043; *p* = 0.218). However, it was positively associated with vegetable consumption (aOR = 1.098; 95%CI: 1.059–1.138; *p* < 0.001).

### 3.3. Participation in Sports

Significant differences were observed between the two groups regarding participation in sports, with non-immigrant children participating at a rate more than twice as high (43.5% vs. 20.9%; *p* < 0.001) compared to immigrant children (43.5% vs. 20.9%; *p* < 0.001).

A logistic regression model was adjusted to identify factors associated with the chance of not participating in sports. The results presented in Table 5 indicate that immigrant children (aOR = 2.185; 95%CI: 1.512–3.158; *p* < 0.001) and children born in 2020 (aOR = 2.496; 95%CI: 1.733–3.595; *p* < 0.001) were twice as likely to not engage in sports compared with non-immigrant children and children born in 2018. Other factors were strongly associated with a higher likelihood of non-participation in sports. Specifically, children from lower-income households and those with mothers with lower educational levels were more likely to not engage in sports compared to those from less disadvantaged backgrounds. As monthly family income and maternal education levels decreased, the odds of children not participating in sports increased.

The length of stay in the host country among immigrant mothers was significantly associated with the child’s participation in sports (aOR = 1.035; 95%CI: 1.004–1.067; *p* = 0.026), after adjusting for family income and the education level of the mother.

## 4. Discussion

Research on migrant child health has been considered a global priority in public health [47]. This paper aims to provide insights into the dietary patterns and sports participation of preschool immigrant children in Portugal and to identify associated factors.

The socioeconomic circumstances in which children grow up play a key role in their health outcomes, including health behaviours [48]. We observed significant socioeconomic inequalities between immigrant and non-immigrant children from an early age. Immigrant children were more likely to live in families with lower incomes, and their mothers experienced higher rates of unemployment and lower levels of education. Limited access to higher education, discrimination in the labour market and precarious employment can be some of the barriers that immigrant families face [49,50]. This profile of disadvantage aligns with the existing literature highlighting income poverty among children from immigrant backgrounds as a challenge in many European countries, including Portugal [51,52]. Structural challenges can increase the socioeconomic vulnerability of immigrant children, impacting long-term health and well-being. Targeted multisectoral policies are essential for improving integration, fostering social and economic inclusion, and reducing health inequalities. Regarding health care access and utilization, our findings are consistent with existing research [53,54,55,56]. Immigrant children were less likely to have an assigned family doctor and attended fewer regular health and development check-ups compared to non-immigrant children. This is concerning because, besides monitoring child health and development, these assessments offer an opportunity to address a range of topics linked to health promotion and disease prevention, including adequate nutrition and physical activity. Administrative, socio-cultural, and language barriers, along with a lack of knowledge about health services in the host country and insufficient human resources in primary health care, may contribute to the lower uptake of routine health assessments. Overcoming these barriers requires cultural mediation when necessary, strengthening primary health care, and a proactive approach to convey information to these families about the availability and importance of health assessments.

Our analysis found that breastfeeding was more prevalent among immigrant mothers. These mothers initiated breastfeeding more frequently, and breastfed exclusively and breastfed in any form significantly longer than mothers of non-immigrant children. These results are consistent with a substantial body of research [17,18,19,20,21,57]; however, a study in Portugal found no differences in initiating breastfeeding between the two groups [20]. Conversely, a study in Denmark concluded that immigrant mothers breastfed their children suboptimally compared with their non-immigrant counterparts [23]. This variation is not surprising, as breastfeeding practices can differ significantly across populations, influenced by socio-cultural and economic contexts and migration-related factors, such as acculturation, that affect both breastfeeding initiation and duration [19,58]. Breastfeeding provides short-term and long-term health, economic, and environmental benefits to children, mothers, and society [59,60]. Our findings suggest that the duration of exclusive breastfeeding remains a challenge for mothers of non-immigrant children. Additionally, it is crucial to promote efforts that help to sustain the breastfeeding practices observed among immigrant mothers. The promotion and support of breastfeeding require a multidimensional universal approach that considers women’s experiences and values, workplace conditions, support systems, and accessible, cultural-diversity-aware health services to facilitate optimal breastfeeding practices for all women.

The analysis conducted to evaluate factors associated with fruit and vegetable intake revealed that immigrant children had 30% and 45% lower odds, respectively, of consuming higher daily portions of fruit and vegetables compared to their non-immigrant counterparts. These findings are consistent with other studies [33,34,37]. Contrasting results were found in studies in Australia and Sweden, where immigrant children had higher odds of meeting fruit and vegetable recommendations [61,62]. Migrant families and their children appear to be more susceptible to food insecurity and micronutrient deficiencies, which can result from a diet lacking in fruits and vegetables. Iron deficiency and low levels of vitamin D are common in migrant children in high-income countries [10,34]. However, it is important to note that vegetable consumption remained a challenge for over 80% of children in our study, who failed to meet the recommended daily intake for this age group. This is especially relevant for preschool children like those in our study (ages 3–5 years), as they are undergoing a period of rapid growth, making them possibly more at risk for iron-deficiency anemia. This condition is associated with increased susceptibility to infections, developmental delays, and behavioural difficulties, which can have long-lasting effects if not identified and addressed early. Hence, it is important to attend yearly regular health and development assessments, as they provide a platform for promoting healthy nutrition and for the early identification of potential concerns. Interestingly, maternal education and family income were not significantly associated with fruit or vegetable intake. This suggests that other factors—such as acculturation, neighbourhood food environments, or availability of culturally appropriate foods—may play a more significant role in shaping dietary behaviours among immigrant children. During the early years, children’s eating habits are shaped by cultural and familial beliefs, with parental influence playing a key role. Yet, it is uncertain whether parent or child nutrition education interventions alone are effective in increasing fruit and vegetable consumption in children under five [63,64]. A more holistic approach that includes families, communities, schools, and economic and social sectors, while considering cultural differences, can effectively promote healthier eating habits. Strategies should include improving access to fresh produce, in which community gardens can have a role in fostering community connections and cultural exchange, together with access to familiar food of diverse communities [65]. Further strategies to combine could involve offering culturally tailored nutrition and education, and supporting school programmes to ensure that all children have access to healthy foods, regardless of background. Adequate intake of fruits and vegetables during early childhood establishes a strong foundation for long-term health. Therefore, initiatives aimed at boosting fruit and vegetable consumption in young children could serve as an effective investment to reduce the overall burden of related diseases later in life.

Participation in organized sports in early life can influence lifelong physical activity patterns, promoting physical health and mental well-being and reducing the risk of developing obesity-related chronic diseases [38]. Our study found that non-immigrant children engaged in an organized sport activity two times more often than immigrant children. The analysis of factors associated with the chance of not participating in sports revealed that an immigrant background, a lower household income, and maternal education increased the odds of not participating in an organized sport activity by two to three times. These results highlight the influence of socioeconomic factors on limiting immigrant children’s opportunities for physical activity, reflecting existing trends in the literature, including the systematic review by Lacoste et al. and two other studies, one in Italy and the other in Denmark [40,41,44]. The low participation in sports among immigrant children has been attributed to socioeconomic barriers, fewer opportunities for access to sports programmes and facilities in immigrant communities, a lack of transportation or language difficulties, and a lack of social integration. Additionally, migrant children from cultures with less emphasis on organized sports may exhibit lower participation rates [42,66]. These findings highlight the need for targeted interventions to promote physical activity among children from disadvantaged backgrounds, including immigrant children. Collaborative partnerships are crucial in addressing this issue. Schools can play a significant role by promoting sports in an inclusive manner and providing accessible facilities. Additionally, local urban planning authorities should focus on creating open spaces for sports, particularly in underserved communities, alongside social policies aimed at improving living standards.

### Limitations and Future Research

While the findings from this study provide insights into the dietary and physical activity behaviours of immigrant and non-immigrant children in Lisbon, there are several limitations to consider. The study’s cross-sectional design limits the ability to draw causal conclusions; future longitudinal studies could provide insights into the long-term effects of these inequalities on child health outcomes. All lifestyle behaviour data were self-reported in our study, which could have introduced recall and social desirability bias. Additionally, the sample was limited to 3–5 year-old children in Portugal, and the findings may not be generalizable to other age ranges or migrant populations. Further research could explore the role of cultural factors, food availability, and acculturation in shaping the dietary and physical activity patterns of immigrant children, as well as conducting detailed analyses of the factors associated with breastfeeding practices.

## 5. Conclusions

This study highlights significant differences in breastfeeding practices, dietary habits, and sports participation between immigrant and non-immigrant children. Breastfeeding practices were better implemented among immigrant children. The findings underscore the need to address social determinants of health and the unique challenges faced by immigrant families in fostering optimal fruit and vegetable consumption and promoting participation in sports among their children. Multisectoral, targeted, culturally appropriate interventions aimed at sustaining good breastfeeding practices, improving access to fruits and vegetables, and increasing sports participation are essential for reducing health inequalities and ensuring that all children have equal opportunities for healthy development.

## Figures and Tables

**Table 1 nutrients-17-01350-t001:** The main sociodemographic characteristics of the children and families.

Variables	Immigrant Children	Non-Immigrant Children	Total	*p* Value
	*n* (%)	*n* (%)	*n* (%)	
	380 (50.0)	380 (50.0)	760 (100)	
**Sex *n* = 760**				*p* = 0.717 *
Female	190 (50.0)	185 (48.7)	375 (49.3)	
**Birth cohort *n* = 760**				*p* < 0.001 *
2018	186 (48.9)	236 (62.1)	422 (55.5)	
2020	194 (51.1)	144 (37.9)	338 (44.5)	
**Mother’s educational level *n* = 759**				*p* = 0.003 **
Less than secondary level	77 (20.3)	87 (22.9)	164 (21.6)	
Secondary level/professional course	192 (50.7)	147 (38.7)	339 (44.7)	
University degree	110 (29.0)	146 (38.4)	256 (33.7)	
**Mother’s employment status *n* = 753**				*p* < 0.001 *
Unemployed	161 (43.0)	113 (29.8)	274 (36.4)	
**Household monthly income *n* = 708**				*p* < 0.001 **
<750 €	121 (34.3)	63 (22.9)	184 (26.0)	
>750–1500 €	160 (45.3)	123 (34.6)	283 (40.0)	
>1500 €	72 (20.4)	169 (47.6)	241 (34.0)	
**Assigned family doctor *n* = 760**				*p* < 0.001 *
Yes	204 (54.1)	308 (81.3)	512 (67.7)	

Significance level: 5%. * Two-proportion z test; ** Fisher–Freeman–Halton Exact Test.

**Table 2 nutrients-17-01350-t002:** Dietary patterns of immigrant and non-immigrant children.

Variables	Immigrant Children	Non-Immigrant Children	Total	*p* Value
	*n* (%)	*n* (%)	*n* (%)	
	380 (50.0)	380 (50.0)	760 (100)	
**Breastfeeding *n* = 759**				*p* < 0.001 *
Yes	365 (96.3)	333 (87.6)	698 (92.0)	
**Duration of exclusive breastfeeding *n* = 693**				*p* < 0.001 ***
Median months (IQ1–IQ3)	6 (3–6)	4 (2–6)	5 (3–6)	
**Total duration of any breastfeeding *n* = 691**				*p* < 0.001 ***
Median months (IQ1–IQ3)	14 (7–24)	8 (3–24)	12 (5–24)	
**Daily portions of fruit *n*= 759**				*p* = 0.026 **
1	48 (12.7)	34 (8.9)	82 (10.8)	
2	154 (40.6)	128 (33.7)	282 (37.2)	
≥3	173 (45.6)	212 (55.8)	385 (50.7)	
None	4 (1.1)	6 (1.6)	10 (1.3)	
**Daily portions of vegetables *n* = 758**				*p* < 0.001 **
1	148 (39.2)	103 (27.1)	251 (33.1)	
2	143 (37.8)	210 (55.3)	353 (46.6)	
≥3	64 (16.9)	56 (14.7)	120 (15.8)	
None	23 (6.1)	11 (2.9)	34 (4.5)	
**Recommended daily intake of fruit ^#^ *n* = 759**				*p* = 0.178 *
Yes (≥2 portions)	327 (86.3)	340 (89.5)	667 (87.9)	
**Recommended daily intake of vegetables ^#^ *n* = 758**				*p* = 0.408 *
Yes (≥3 portions)	64 (16.9)	56 (14.7)	120 (15.8)	

Significance level: 5%. * Two-proportion *z* test; ** Fisher–Freeman–Halton Exact Test; *** Mann–Whitney U Test. ^#^ According to the recommended servings/day by the Portuguese Health Directorate [28].

**Table 3 nutrients-17-01350-t003:** Factors associated with likelihood of consuming ≥ 3 portions/day of fruit.

Variables	aOR	95%CI	*p* Value
**Birth cohort year**			
2018	1.039	0.767–1.407	0.807
2020	reference		
**Immigrant status**			
Non-immigrant child	reference		
Immigrant child	0.700	0.511–0.959	0.027
**Household monthly income**			
≤750 €	reference		
>750–1500 €	1.358	0.928–1.988	0.115
>1500 €	1.490	0.959–2.315	0.076
**Mother’s educational Level**			
Less than secondary	reference		
Secondary	0.800	0.539–1.188	0.269
University degree	0.537	0.552–1363	0.537

**Table 4 nutrients-17-01350-t004:** Factors associated with likelihood of consuming ≥ 2 portions/day of vegetables.

Variables	aOR	95%CI	*p* Value
**Birth cohort year**			
2018	0.550	0.398–0.761	<0.001
2020	reference		
**Immigrant status**			
Non-immigrant child	reference		
Immigrant child	0.489	0.350–0.684	<0.001
**Household monthly income**			
≤750 €	reference		
>750–1500 €	1.163	0.785–1.722	0.452
>1500 €	0.990	0.627–1.562	0.964
**Mother’s educational level**			
Less than secondary	reference		
Secondary	1.101	0.732–1.655	0.643
University degree	1.609	0.999–2.591	0.050

**Table 5 nutrients-17-01350-t005:** Factors associated with likelihood of non-participation in sport.

Variables	aOR	95%CI	*p* Value
**Birth cohort year**			
2018	reference		
2020	2.496	1.733–3.595	<0.001
**Immigrant status**			
Non-immigrant child	reference		
Immigrant child	2.185	1.512–3.158	<0.001
**Household monthly income**			
≤750 €	3.181	1.861–5.437	<0.001
>750–1500 €	1.761	1.165–2.662	0.007
>1500 €	reference		
**Mother’s educational level**			
Less than secondary	3.555	2.036–6.206	<0.001
Secondary	2.087	0.732–1.655	<0.001
University degree	reference		

## Data Availability

The data presented in this study are available on request from MROM.

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
