# Peer review of "Breastfeeding Duration, Diet, and Sports Engagement in Immigrant Children: A Quantitative Study in the Lisbon Region, Portugal"

_nutrients, 2025, doi:10.3390/nu17081350_

Round 1

Reviewer 1 Report

Comments and Suggestions for Authors

The topic is quite surprising and touches on a lot of rather oddly chosen elements. I don't think that combining such divergent areas is appropriate in one manuscript.

My comments:

  1. Line 12: "While immigrant children often exhibit better breastfeeding practices," - this sentence surprises me quite a bit - it seems to me that this is not a very successful translation - it should be reworded
  2. line 28/29 - "Being an immigrant child decreased significantly the odds of consuming three or more portions of fruit" - also probably a failed translation
  3. the conclusions in the abstract seem a bit surprising to me - especially in relation to breastfeeding, which was much better implemented in immigrant families
  4. introduction - no comments
  5. I am surprised that the groups are so evenly matched - it is not impossible, but very unlikely
  6. methods - no comments
  7. results:
  • Table 2 - explain why the numbers of respondents for the individual parameters analysed are different - whether some answers were excluded or some respondents did not provide an answer.
  • Table 3 and Table 4 - n=757 - why not 760 (there were probably that many respondents - three are missing - why?)
  1. Discussion - consider shortening the introduction to the discussion of your results, it is not strictly related to the topic of the manuscript and unnecessarily prolongs the discussion, which is really too long .
  2. Conclusions - in my opinion the conclusion about breastfeeding (as I wrote above) is inconsistent with the results. The rest is ok
  3. improve reference formatting according to MDPI guidelines
Comments on the Quality of English Language

The translation of some fragments is not very successful and may mislead the reader.

Author Response

Thank you very much for taking the time to review this manuscript. Please find a point-by-point response below and the corresponding revisions in track changes in the re-submitted file. 

  1. Line 12: "While immigrant children often exhibit better breastfeeding practices," - this sentence surprises me quite a bit - it seems to me that this is not a very successful translation - it should be reworded

Thank you for this comment. We decided to remove it, instead, we added a sentence that contextualises these areas in shaping long term well-being:  "Being breastfed, following a healthy diet and staying active during childhood shape health trajectories across the life course, promoting long-term well-being. "  (lines 12-13)

     2.  line 28/29 - "Being an immigrant child decreased significantly the odds of consuming three or more portions of fruit" - also probably a failed translation

We reworded the sentence to ” Immigrant children had significantly lower odds of consuming three or more portions of fruit” ( lines 33-33)

    3.  the conclusions in the abstract seem a bit surprising to me - especially in relation to breastfeeding, which was much better implemented in immigrant families

Thank you for pointing this out . Our results show indeed that breastfeeding was better implemented among immigrant families. However, it is also our understanding , that it is essential to focus on sustaining these practices, to ensure their long-term success. We have reworded the sentence (line 38 ) to emphasize this.

    4.  introduction - no comments

    5.  I am surprised that the groups are so evenly matched - it is not impossible, but very unlikely

Please refer to the methods section (lines 142-43): "one non-immigrant child was matched to one immigrant to overcome possible lost to follow-up of immigrants."

    6.  methods - no comments

    7.  results:

Table 2 - explain why the numbers of respondents for the individual parameters analysed are different - whether some answers were excluded or some respondents did not provide an answer.

The numbers are different because respondents did not provide an answer. This information was added to lines 190-191

Table 3 and Table 4 - n=757 - why not 760 (there were probably that many respondents - three are missing - why?)

Thank you for this comment . This was a mistake and we removed it.

   8.  Discussion - consider shortening the introduction to the discussion of your results, it is not strictly related to the topic of the manuscript and unnecessarily prolongs the discussion, which is really too long .

We agree. Introduction and other sections of the discussion have been shortened.

   9.  Conclusions - in my opinion the conclusion about breastfeeding (as I wrote above) is inconsistent with the results. The rest is ok

Thank you for your comment. We clarified above on comment 3 and reworded the statements related to breastfeeding on conclusions: " Breastfeeding practices were better implemented among immigrant children.......................  Multisectoral, targeted, culturally appropriate interventions aimed at  sustaining good breastfeeding practices,................" (lines 456 and 460-61)

    10.  improve reference formatting according to MDPI guidelines

Thank you for pointing this out . Reference formatting was redone as per MDPI guidelines (lines 493-701)

Reviewer 2 Report

Comments and Suggestions for Authors

Please indicate whether the data concern any breastfeeding or exclusive breastfeeding.

Have the authors attempted to further analyze the data to look for the relationship between the breastfeeding and unemployment (work regime and distance to the workplace can affect the breastfeeding habits) and country of origin. The paper of Kana et al. is cited; that study revealed that maternal country of birth does not influence breastfeeding initiation and type of feeding practice. However, data of Kana et al. point to differences between South American and African immigrants as far as any breastfeeding is concerned; moreover, there may be differences between various cohorts studied (the conclusion of Kana et al. could be confirmed by new results or discussed on their basis).

Were mixed Portuguese/immigrant couples analyzed and, if so, were there any differences between fully immigrant and mixed couples?

Were there any exclusion criteria (mother illness, lactose intolerance etc.) which interfered with the breastfeeding?

There is a discrepancy between the Abstract and the text: the Abstract states that “Approximately 720 children” were enrolled in the study while in fact, 760 were analyzed.

References should be formatted according to the journal requirements (they are in the Pubmed style).

Author Response

Thank you very much for taking the time to review this manuscript. Please find a point-by-point response below and the corresponding revisions in track changes in the re-submitted file.

  1. Please indicate whether the data concern any breastfeeding or exclusive breastfeeding.

Thank you for pointing this out. The data concerns any breastfeeding in the variable total duration of breastfeeding and when exclusive breastfeeding was being analysed this was explicit. We have now clarified this along the text ( lines 175,235,353) and in table 2 .

    2.  Have the authors attempted to further analyze the data to look for the relationship between the breastfeeding and unemployment(work regime and distance to the workplace can affect the breastfeeding habits) and country of origin.  

We found that unemployed mothers had 1.4 more odds of completing 6 months of exclusive breastfeeding than their employed counterparts, adjusting for mothers education and migrant status ( aOR=1.39;95% CI:1.013-1.912; p=0.041). However, we did not include this information in the current paper because we plan to conduct a separate study focused exclusively on breastfeeding practices among immigrant mothers, where we will perform more detailed analyses. This is addressed in the 'Further Research' section ( lines 447-451)

    3.  The paper of Kana et al. is cited; that study revealed that maternal country of birth does not influence breastfeeding initiation and type of feeding practice. However, data of Kana et al. point to differences between South American and African immigrants as far as any breastfeeding is concerned; moreover, there may be differences between various cohorts studied (the conclusion of Kana et al. could be confirmed by new results or discussed on their basis).Were mixed Portuguese/immigrant couples analyzed and, if so, were there any differences between fully immigrant and mixed couples?

Please refer to response to comment 2.

    4.  Were there any exclusion criteria (mother illness, lactose intolerance etc.) which interfered with the breastfeeding?

Thank you for this question. Our database is from a cohort study not designed to focus on breastfeeding and information on the reasons which could have interfered  with this practice was not collected . 

   5.  There is a discrepancy between the Abstract and the text: the Abstract states that “Approximately 720 children” were enrolled in the study while in fact, 760 were analyzed.

Thank you very much for pointing this out - it was a mistake. This as now been corrected to 760  ( line 20)

    6.  References should be formatted according to the journal requirements (they are in the Pubmed style).

Thank you. We have  now formatted the references  as per journal requirements( lines 493-701)

Reviewer 3 Report

Comments and Suggestions for Authors
  • Title: You can delete „children‘s” in front of “diet”, because this is already mentioned later on.
  • Line 25 and others: Please correct into “95% CI”.
  • Introduction: The section on breastfeeding is very long – particularly compared to physical activity.
  • Introduction: Please note that physical activity is not equal to organized sports. This needs further differentiation and elaboration.
  • Line 117: What is meant with “first wave”? those included in 2018?
  • Line 120: Why do you refer to 2022?
  • Line 160: Why is breastfeeding only a secondary outcome?
  • Lines 173-174: the equal numbers of migrants and non-migrants is due to the matching procedure.
  • Table 1: Why didn`t you take sex and birth cohort into account during matching?
  • Lines 193: Please correct, because this is not two times.
  • Line 211: Please rephrase, because it is not the children who initiated breastfeeding.
  • In the discussion section, you refer to often to single studies. Please try to summarize/synthesize a bit more.

Author Response

Thank you very much for taking the time to review this manuscript. Please find a point-by-point response below and the corresponding revisions in track changes in the re-submitted file.

  1. Title: You can delete „children‘s” in front of “diet”, because this is already mentioned later on.

We agree, it is now deleted.

    2.   Line 25 and others: Please correct into “95% CI”.

Thank you for pointing this out. It was all corrected to 95% CI.

    3.  Introduction: The section on breastfeeding is very long – particularly compared to physical activity.

Thank you for your comment.  We shorten the section (lines 61-85).

    4.   Introduction: Please note that physical activity is not equal to organized sports. This needs further differentiation and elaboration.

Thank you for pointing this out. We have now indicated that organized sports contribute to increased levels of physical activity (lines 113-14) and we have rephrased the aims of the paper using participation in sports instead of physical activity ( line 121). This has also been addressed in the main outcomes section 2.4.2 ( lines 167-8; 171-2)

    5.   Line 117: What is meant with “first wave”? those included in 2018?

We meant we used data from the first data collection for each age cohort. This clarification was added to lines 127-8 " we used data from the first wave of data collection from each age cohort"

   6.   Line 120: Why do you refer to 2022?

Because children started to be enrolled in the study in May 2022; please refer to  the recruitment section 2.3 :"Recruitment took place sequentially, with children born in 2018 and 2020 enrolled between May 2022 and April 2024"(lines 150-1)

   7.   Line 160: Why is breastfeeding only a secondary outcome?

Our main objectives in this study are related to diet and sports engagement.However, we thought it would also be interesting to analyse data on breastfeeding in this context. We therefore considered this variable as a secondary outcome, which will be analysed in more detail in another article.

   8.  Lines 173-174: the equal numbers of migrants and non-migrants is due to the matching procedure.

We have added this info on lines 189-190 : "There was an equal distribution between non-immigrant (n=380) and immigrant children (n=380) due to the matching procedure"

   9.  Why didn`t you take sex and birth cohort into account during matching?

The question you raise is very pertinent. Our option for these two variables was not to match and always include them in the adjusted models. This is easy to implement in practice, where recruitment, especially for migrant children, is already complex. This seemed to be the better option given the time available to complete the study.

   10.  Lines 193: Please correct, because this is not two times.

Thank you for pointing this out. We have corrected in line 192: " Immigrant families (34.3%) were one and a half times......."

   11.   Line 211: Please rephrase, because it is not the children who initiated breastfeeding.

Agree. Sentence was rephrased to “mothers of immigrant children were not only more likely to initiate breastfeeding….”( line 232 )

   12.    In the discussion section, you refer to often to single studies. Please try to summarize/synthesize a bit more.

Thank you for your comment.  We have synthesized the discussion overall, aiming to offer a more engaging read.

Round 2

Reviewer 1 Report

Comments and Suggestions for Authors

The authors have made satisfactory improvements that increase the substantive value of this manuscript. I have no other comments.

Comments on the Quality of English Language

The text, in my opinion, requires linguistic correction. Some translations are not very successful.

Reviewer 3 Report

Comments and Suggestions for Authors

All previous comments have adequately been addressed.